# Adaptive Semantic Chunking Method for Medical Retrieval-Augmented Generation Systems

E. Fartushny, G. Lebedev, V. Kuzovatova

January 2026

**Abstract**

Retrieval-augmented generation (RAG) systems are becoming a key tool for handling medical documentation. However, retrieval quality critically depends on the document splitting strategy (chunking). Existing methods—fixed window, sliding window, semantic splitting—do not account for the specifics of clinical guidelines: hierarchical structure, the "recommendation—evidence level" link, medical terminology.

This paper proposes the ASCM method (Adaptive Semantic Chunking Method)—an adaptive chunking algorithm with proven optimality. The task is formalized as finding the shortest path in a directed acyclic graph (DAG). A multi-component coherence function combines semantic, structural, and medical analysis. The dynamic programming algorithm guarantees global optimum in time $O(N \cdot L_{\max})$.

Experiments on a corpus of gastrointestinal oncology clinical guidelines show superiority of ASCM over baseline methods: +18% in medical entity preservation rate (EPR), +12% in semantic coherence score (SCS).

Keywords: Generative language models, LLM, RAG, chunking, semantic search, clinical guidelines, evidence-based medicine, NLP in medicine

# 1 Introduction

## 1.1 Relevance

Retrieval-augmented generation (RAG) systems are actively being introduced into medical practice for automating work with clinical guidelines, treatment protocols, and scientific literature [1]. A key stage in building RAG is splitting documents into fragments (chunks), which are indexed for subsequent search.

Chunking quality directly affects:

- Search precision — relevance of found fragments to the query

- Context completeness — sufficiency of information for answer generation

- Correctness — absence of breaks in logically connected entities

## 1.2 Problem

Existing chunking methods were developed for general texts and do not account for the specifics of medical documents:

1. Hierarchical structure of clinical guidelines: sections → subsections → recommendations → comments

2. Evidence-based medicine: inseparable link "Recommendation + Level of Conviction (LoC) + Level of Reliability (LoR)"

3. Medical terminology: complex compound terms (TNM staging, drug names, ICD-10 diagnoses)

4. Tables and algorithms: chemotherapy schemes, diagnostic algorithms

## 1.3 Contribution

1. Formalization of the chunking task as a combinatorial optimization problem on a graph

2. Multi-component coherence function taking into account semantics, structure, and medical constraints

3. DP-algorithm with proven optimality and complexity $O(N \cdot L_{\max})$

4. Experimental validation on a corpus of oncology clinical guidelines

# 2 Literature Review

## 2.1 General-Purpose Chunking Methods

Fixed-Size Chunking [2] — splitting into fixed-length blocks. Implemented in most frameworks (LangChain, Haystack). Advantage: simplicity. Disadvantage: ignores semantic boundaries.

Recursive Character Text Splitter [3] — hierarchical splitting by delimiters (paragraphs → sentences → characters). Advantage: accounts for structure. Disadvantage: fixed rules without adaptation.

Semantic Chunking [4] — splitting based on drop in cosine similarity of embeddings between adjacent sentences. Implemented in LlamaIndex. Advantage: accounts for semantics. Disadvantage: greedy algorithm without optimality guarantee.

## 2.2 Chunking for Medical Texts

Chen et al. (2023) [5] proposed using UMLS for boundary detection, but without formal optimization.

MedRAG (Stanford, 2024) [6] applies fixed chunks of 512 tokens with dense retrieval, showing limitations on complex queries.

BioGPT [7] uses standard sliding window without adaptation to document structure.

## 2.3 Research Gap

None of the existing methods provide:

- Mathematically justified optimization

- Accounting for specifics of clinical guidelines (LoC/LoR)

- Hard constraints on medical entity preservation

# 3 ASCM Method

## 3.1 Problem Statement

Let document $D$ be represented as a sequence of sentences $S = \{s_1, s_2, \ldots, s_N\}$. Task: find a partition $P^* = \{Ch_1, \ldots, Ch_M\}$, minimizing the global cost function:

$$P^* = \arg\min_P \sum_{k=1}^{M} \Phi(Ch_k) \qquad (1)$$

under constraints:

- $L_{\min} \leq |Ch_k| \leq L_{\max}$ (soft)

- Prohibition of breaks within protected entities (hard)

## 3.2 Transition Graph

Construct DAG $G = (V, E)$:

- $V = \{0, 1, \ldots, N\}$ — positions between sentences

- $(i, j) \in E$ if $L_{\min} \leq j - i \leq L_{\max}$ and transition is allowed

- $w(i, j) = \Phi(i, j)$ — cost of chunk $\{s_{i+1}, \ldots, s_j\}$

## 3.3 Cost Function

$$\Phi(i, j) = \frac{1}{1 + C_{\text{boundary}}(j)} + \lambda \cdot P_{\text{len}}(j - i) + \mu \cdot P_{\text{coh}}(i, j) \qquad (2)$$

where:

- $C_{\text{boundary}}(j)$ — coherence at boundary

- $P_{\text{len}}$ — penalty for deviation from optimal size

- $P_{\text{coh}}$ — penalty for low internal connectivity

## 3.4 Multi-Component Coherence

Table 1: Components of coherence

| Component | Description | Computation |
|---|---|---|
| $C_{sem}$ | Semantic similarity | $\cos(\mathrm{emb}(s_i), \mathrm{emb}(s_{i+1}))$ |
| $C_{struct}$ | Structural relation | 1 if same section, 0 otherwise |
| $C_{med}$ | Medical constraint | 1 if inseparable pair |

$$C_{\text{total}}(s_i, s_{i+1}) = \alpha \cdot C_{sem} + \beta \cdot C_{struct} + \gamma \cdot C_{med} \qquad (3)$$

Medical constraints ($C_{med} = 1$):

- Break of UMLS/SNOMED entity

- Break of pair "Recommendation — LoC/LoR"

- Break of numbered list elements

## 3.5 Dynamic Programming Algorithm

---
**Algorithm 1** ASCM($S$, $L_{\min}$, $L_{\max}$, $\alpha$, $\beta$, $\gamma$, $\lambda$, $\mu$)
---
1: $N \leftarrow |S|$
2: $dp[0] \leftarrow 0$, $parent[0] \leftarrow -1$
3: **for** $i \leftarrow 1$ to $N - 1$ **do**
4: $\quad C[i] \leftarrow \alpha \cdot C_{sem}(i) + \beta \cdot C_{struct}(i) + \gamma \cdot C_{med}(i)$
5: **end for**
6: $forbidden \leftarrow \{i : C_{med}(i) = 1\}$
7: **for** $j \leftarrow 1$ to $N$ **do**
8: $\quad dp[j] \leftarrow \infty$
9: $\quad$ **for** $i \leftarrow \max(0, j - L_{\max})$ to $j - L_{\min}$ **do**
10: $\quad\quad$ **if** valid_transition($i$, $j$, $forbidden$) **then**
11: $\quad\quad\quad cost \leftarrow dp[i] + \Phi(i, j)$
12: $\quad\quad\quad$ **if** $cost < dp[j]$ **then**
13: $\quad\quad\quad\quad dp[j] \leftarrow cost$
14: $\quad\quad\quad\quad parent[j] \leftarrow i$
15: $\quad\quad\quad$ **end if**
16: $\quad\quad$ **end if**
17: $\quad$ **end for**
18: **end for**
19: **return** reconstruct_path($parent$, $N$)

---

**Theorem 1 (Optimality)** *Algorithm ASCM finds the globally optimal partition.*

The task is equivalent to finding the shortest path in a DAG. The DP-algorithm traverses vertices in topological order, which guarantees optimality [8].

Complexity: $O(N \cdot L_{\max})$ time, $O(N)$ memory.

## 3.6 Algorithm Choice Justification

### 3.6.1 Alternative Optimization Approaches

Table 2: Comparison of optimization algorithms

| Algorithm | Optimality Guarantee | Complexity | Applicability |
|---|---|---|---|
| Greedy | No | $O(N)$ | Prototypes, LlamaIndex |
| Beam Search | No | $O(N \cdot k \cdot L_{\max})$ | Speed/quality trade-off |
| A* | Yes* | $O(N \cdot L_{\max}) - O(N^2)$ | With good heuristic |
| ILP | Yes | Exponential (worst) | Complex global constraints |
| Genetic Algorithm (GA) | No | Configurable | Non-differentiable functions |
| Reinforcement Learning | No | $O(N)$ + training | With labeled data |
| DP (our choice) | Yes | $O(N \cdot L_{\max})$ | Optimal for DAG structure |

### 3.6.2 Why Dynamic Programming?

1. Optimal substructure: if $P^* = \{Ch_1, \ldots, Ch_M\}$ is the optimal partition of $S[1:N]$, then $\{Ch_1, \ldots, Ch_{M-1}\}$ is the optimal partition of $S[1:b_{M-1}]$

2. Overlapping subproblems: computing $dp[j]$ for different paths uses the same $dp[i]$ values

3. DAG structure: transition graph is acyclic by construction ($i < j$ for all edges), guaranteeing correct topological traversal

4. Determinism: critical for medical applications — reproducibility of results

### 3.6.3 Experimental Algorithm Comparison

Table 3: Performance of algorithms on 1K sentences

| Algorithm | SCS | EPR | ERL | Time |
|---|---|---|---|---|
| Greedy | 0.71 | 0.81 | 0.58 | 0.02s |
| Beam-10 | 0.75 | 0.88 | 0.72 | 0.18s |
| Beam-50 | 0.77 | 0.91 | 0.81 | 0.85s |
| DP (ASCM) | 0.79 | 0.95 | 0.91 | 0.08s |

# 4 Experiments

## 4.1 Dataset

Corpus of 15 clinical guidelines on gastrointestinal oncology (gastric cancer, colorectal cancer, GIST). Total volume: 180,000 tokens, 3,200 sentences.

## 4.2 Baseline Methods

- Fixed-512: Fixed chunks of 512 tokens

- LangChain: RecursiveCharacterTextSplitter

- LlamaIndex: SemanticSplitterNodeParser

- ASCM: Proposed method

## 4.3 Metrics

1. SCS (Semantic Coherence Score) — average intra-chunk cosine similarity

2. EPR (Entity Preservation Rate) — proportion of unbroken medical entities

3. ERL (Evidence-Recommendation Linking) — proportion of preserved "recommendation — LoR" pairs

## 4.4 Results

Table 4: Results of chunking methods

| Method | SCS ↑ | EPR ↑ | ERL ↑ | Avg Size |
|--------|-------|-------|-------|----------|
| Fixed-512 | 0.61 | 0.72 | 0.45 | 512 |
| LangChain | 0.64 | 0.78 | 0.52 | 480 |
| LlamaIndex | 0.71 | 0.81 | 0.58 | 520 |
| ASCM | 0.79 | 0.95 | 0.91 | 580 |

Improvements of ASCM vs best baseline:

- SCS: +11% (0.79 vs 0.71)

- EPR: +17% (0.95 vs 0.81)

- ERL: +57% (0.91 vs 0.58)

### 4.5 Examples

Example of break in LangChain:
Chunk 1: "D2 lymph node dissection is recommended for gastric cancer"
Chunk 2: "(LoC A, LoR 1). Comment: D2 lymph node dissection..."

ASCM preserves the link:
Chunk 1: "D2 lymph node dissection is recommended for gastric cancer (LoC A, LoR 1). Comment: D2 lymph node dissection ensures..."

# 5 Discussion

## 5.1 Advantages of ASCM

1. Optimality guarantee — unlike greedy heuristics

2. Modularity — weights $\alpha$, $\beta$, $\gamma$ tunable per domain

3. Extensibility — easy to add new coherence components

4. Efficiency — practically linear complexity

## 5.2 Limitations

1. Requires preliminary document structure annotation

2. Dependence on embedding model quality for $C_{sem}$

3. Need for hyperparameter tuning for specific domain

# 6 Conclusion

The ASCM method is proposed — the first chunking algorithm for medical RAG systems with mathematically proven optimality.

Formalization of the task as a shortest path problem on a DAG allows applying an efficient DP-algorithm.

Key innovation — multi-component coherence function combining semantic analysis, document structure, and domain-specific constraints (medical terms, UMLS entities, links in clinical guidelines and evidence-based medicine "recommendation — evidence — level").

Experiments on a corpus of oncology clinical guidelines demonstrate significant superiority over existing methods: +17% in medical entity preservation, +57% in recommendation-evidence link preservation.

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
