# OpenReview forum: "Adaptive Semantic Chunking Method for Medical Retrieval-Augmented Generation Systems"
_mathai.club/MathAI/2026/Conference — Submitted to 2026_

### Official Review · Reviewer_16NR · 2026-03-11
**Promising approach that lacks of experiments breakdown**

**Rating:** 3
**Confidence:** 4

**Review:**

This paper proposes an adaptive chunking algorithm for medical documents in order to be used in RAG/agent systems.


**Strengths**

1. The focus of the work on real-world scenarios and practical results - the proposed method can increase quality of agent systems in medicine which is a sensitive domain.
2. Mathematical explanations of the proposed method - authors provided an equations and pseudo-code that can help deeply understand the core idea of the approach.

**Weaknesses**

1. Paper is not anonymized.
2. Paper doesn't meet the conference style.
3. The existing chunking approaches might be explained in more detail.
4. Authors declaring the strong metrics improvements over baselines, but the experimental setup and performed steps wasn't described at all, only raw enumerations provided.
5. The source code of the method and the experiments will marginally increase the quality of the paper.

---

### Decision · Program_Chairs · 2026-03-14

**Decision:**

Reject

**Comment:**

After careful evaluation by the Program Committee, we regret to inform you that your submission has not been accepted for presentation at MathAI 2026.

All submissions underwent a rigorous two-stage review process. Unfortunately, the reviewers identified one or more of the following concerns with your paper:

- Insufficient mathematical rigor or novelty relative to the existing body of work in the field;
- Presentation of results that substantially overlap with or rephrase previously published findings without clear original contribution;
- Significant issues with technical quality, including but not limited to broken or non-existent references, unsupported claims, or methodological gaps;
- Indications that the manuscript may have been generated with the assistance of large language models without substantial original intellectual contribution by the authors.

We received a large number of submissions this year, and the selection process was highly competitive. We encourage you to carefully consider the reviewers’ feedback (available through OpenReview), revise your work accordingly, and consider submitting an improved version to a future edition of MathAI or to another appropriate venue.

We appreciate your interest in MathAI and hope you will continue to engage with the conference community.

With kind regards,

MathAI 2026 Program Committee
URL: https://mathai.club
Telegram: https://t.me/MathAI_club
Email: mathai.club@yandex.ru